# Adsorption in Mixtures with Competing Interactions

**DOI:** 10.3390/molecules26154532

**Published:** 2021-07-27

**Authors:** Marek Litniewski, Alina Ciach

**Affiliations:** Institute of Physical Chemistry, Polish Academy of Sciences, 01-224 Warsaw, Poland; mlitniewski@ichf.edu.pl

**Keywords:** adsorption, mixture of charged particles, competing interactions, self-assembly

## Abstract

A binary mixture of oppositely charged particles with additional short-range attraction between like particles and short-range repulsion between different ones in the neighborhood of a substrate preferentially adsorbing the first component is studied by molecular dynamics simulations. The studied thermodynamic states correspond to an approach to the gas–crystal coexistence. Dependence of the near-surface structure, adsorption and selective adsorption on the strength of the wall–particle interactions and the gas density is determined. We find that alternating layers or bilayers of particles of the two components are formed, but the number of the adsorbed layers, their orientation and the ordered patterns formed inside these layers could be quite different for different substrates and gas density. Different structures are associated with different numbers of adsorbed layers, and for strong attraction the thickness of the adsorbed film can be as large as seven particle diameters. In all cases, similar amount of particles of the two components is adsorbed, because of the long-range attraction between different particles.

## 1. Introduction

Phenomena associated with adsorption on solid surfaces have been studied for decades for systems ranging from noble gases to liquid crystals and polymers [1,2,3,4]. Still, adsorption in systems with competing interactions, such as short-range attraction and long-range repulsion (SALR) [5,6,7,8,9,10,11,12], have attracted attention only recently [13,14]. The long-range repulsion between the particles originates typically from screened electrostatic interactions, and the attraction is often mediated by complex solvents. In particular, SALR-type interactions were observed for globular lysozyme particles in deionized water and between colloid particles in a solvent containing polymers with a radius of gyration much smaller than the radius of the particles [5,6]. The competing interactions in the SALR systems can lead to spontaneously formed stable aggregates of particles, such as spherical or elongated clusters, networks or layers. The size of the spherical clusters is determined by the range of the attraction, and the distance between them by the range of the repulsion, unless the density is too large and more complex structures must be formed [15,16].

The clusters start to be formed when the volume fraction of the particles exceeds the value known as the critical cluster concentration [17], analogous to the critical micelle concentration [18]. The number of clusters increases with the increasing volume fraction of the particles. At the crossover between monomer dominated- and cluster-dominated systems, the probabilities that a randomly chosen particle is isolated or belongs to a cluster of the optimal size are equal. At this crossover, the specific heat takes a maximum [13]. It turned out that the adsorption as a function of the chemical potential takes a maximum at this crossover as well [14], and decreases for increasing chemical potential in the cluster-dominated system. This anomalous behavior of adsorption originates from repulsion between the clusters. At an attractive surface, a layer of clusters separated by distances larger than the range of the repulsion is formed. In order to avoid the repulsion, the adsorbed clusters form a local hexagonal pattern. New clusters are repelled by the adsorbed ones, and the adsorbed repulsive layer ’screens’ the attractive surface. As a result, a depletion zone is formed after the adsorbed layer of particles [13]. These results indicate that aggregation of particles has a very significant effect on their adsorption on solid substrates, and that adsorption in self-assembling systems deserves much more attention.

Self-assembly into different aggregates is ubiquitous in biological and soft-matter systems that typically consist of a mixture of charged macromolecules or nanoparticles in solvents composed of much smaller molecules. In mixtures, one can expect formation of different patterns, depending on the types of interactions and thermodynamic state. Rich variety of ordered structures (colloidal crystals) was found in binary mixtures of oppositely charged colloid particles in solvents with properly tuned dielectric constant [19,20]. In molten colloidal crystals or fluid-like phases, local order can still be present. Local ordering in a binary mixture with an energetically favored neighborhood of particles of different components was studied by theory and simulation [21]. However, mixtures of particles with competing interactions leading to the formation of clusters of particles of the same kind were studied much less intensively [22,23]. To the best of our knowledge, effects of self-assembly in such mixtures on the adsorption have not been studied at all. The effect of a presence of the second component on the adsorption of the first-component particles is an entirely open question that we address in this work.

In this work we focus on adsorption phenomena in a mixture of particles with a particular form of competing interactions. We choose the system for which the phase diagram was determined by theory and simulations in [23]. In the binary mixture studied in [23], spherical particles with equal diameter and SALR interactions between particles of the same kind were considered. Particles of different kind repel and attract each other at short and large separations, respectively. Such long-range interactions can come from the screened electrostatic forces, when the particles of different species are oppositely charged. The short-range attraction between like particles and short-range repulsion between different ones can result from the thermodynamic Casimir potential [24], if its temperature-controlled range is shorter than the range of the screened Coulomb potential. Attractive and repulsive Casimir interactions between like and different surfaces, respectively, are induced by concentration fluctuations in a binary solvent close to a critical point of the demixing phase transition, and were studied by theory and experiment [24,25,26]. The range of the Casimir interactions is determined by the correlation length of the concentration fluctuations in the solvent. Since this length increases when the critical temperature is approached and depends sensitively on temperature, the Casimir interactions can be precisely controlled. In [27,28] static and dynamic properties of paramangentic particles in magnetic field suspended in a critical water–lutidine mixture were studied by density functional theory. The combined Casimir attraction and long-range dipole–dipole repulsion was suggested to be particularly suitable for experimental observation of the ordered periodic phases in the SALR systems. Recent experiments have shown that quantum dots interacting with Casimir and screened electrostatic potentials self-assemble into aggregates that agree well with simulation results [29], suggesting that such systems can indeed be used in future experimental studies of self-assembling particles. We should note, however, that despite theoretical and simulation predictions, the phases with long-range order have not been observed experimentally yet [8]. Our theoretical model is motivated by the Casimir and screened electrostatic potentials, but this type of interactions in binary mixtures may be present in different complex systems as well.

The phase diagram of the binary mixture with competing interactions was determined in [23] for equal densities of the two components. When the number of particles of the two components is equal, the mixture undergoes a transition into a gas and a dense phase with alternating layers rich in the first and the second component. At low temperature, the layers have a crystalline structure. Crystalline structure in the aggregates was also observed at low *T* in one-component SALR systems both in simulations and experiments [29,30]. The crystal melts at higher temperature, but alternating liquid-like layers rich in the first and the second component remain present up to a temperature at which a transition to a disordered isotropic phase takes place. The theoretically predicted phase diagram has not been confirmed experimentally yet, since the studies of the self-assembly of particles interacting with the Casimir and screened electrostatic potentials were limited to one-component systems [29,31]. We hope that our predictions will motivate future experimental studies.

Here we focus on the question if the adsorption at a selective surface attracting only the first component will be altered by the presence of the second component. Particles of different species attract each other at large distances, and one can expect that the adsorbed layer of particles of the first kind can attract particles of the second kind, and so on. This way the depletion zone found in the one-component SALR system could be replaced by a layer of particles of the second component. As we show in the following sections, much more interesting behavior, with patterns depending sensitively on the interaction with the substrate can be formed.

In Section 2 we introduce the model and describe briefly the molecular dynamics (MD) simulation method. The results of simulations are presented in Section 3. In Section 3.1, distribution of particles and patterns formed near the substrate are described, and in Section 3.2 we present the adsorption and the selective adsorption, i.e., near-surface excess of the first component over the second one, as functions of the gas density and the strength of the wall–particle attraction. Section 4 contains summary and discussion.

## 2. The Model and the Simulation Method

### 2.1. The Model

We consider spherical particles with equal diameters, and assume the same interactions as in [23]. The interaction potential between particles of the same kind is given by
(1)$$u_{ii}\left(r\right)=\frac{6\epsilon }{r^{12}}-\frac{6\epsilon }{r^{6}}+1.8\frac{e^{-r/2}}{r},$$
where i=1,2. Such interactions were assumed in studies of adsorption in the one-component SALR system [13]. The parameters in the interaction potential lead to relatively small range of attraction, and therefore small clusters composed of several particles are formed. This allows for a large number of clusters for a reasonable number of particles in our simulations. The interactions between different particles are given by
(2)$$u_{12}\left(r\right)=\frac{6\epsilon }{r^{12}}+\frac{6\epsilon }{r^{6}}-1.8\frac{e^{-r/2}}{r}.$$

For $r=r_{c}=6.75$ the potentials are truncated. The energy unit is given by ϵ, and the length is measured in units of the particle diameter *a*. Simulation results for the bulk phase diagram agreed on a semiquantitative level with theoretical results obtained for particles with strictly hard cores ($u_{ij}\left(r\right)=\infty$ for r<1) [23]. Thus, we can expect that the model is suitable for different types of hard particles with smooth cores. Because of the very strong repulsion for r<1, however, the model is not suitable for soft particles with cores that can overlap.

The diameter of quantum dots, nanoparticles or colloid particles for which the SALR interactions are observed, ranges from a few to a few hundreds of nanometers. On such a large length scale, the microscopic structure of the solid substrate does not play a significant role, and we assume an ideal, flat surface parallel to the (x,y) plane and placed at z=0, with the particle–substrate interactions depending only on the distance *z* from the substrate. We assume that the interactions of the first and the second component with the solid substrate have the same form as in [23], but the strength is controlled by the parameters $\gamma_{attr}$ and $\gamma_{rep}$, respectively, and for z>0,
(3)$$V_{1}^{wall}\left(z\right)=\gamma_{attr}\left(\frac{2}{z^{12}}-\frac{2}{z^{6}}\right),$$
and
(4)$$V_{2}^{wall}\left(z\right)=\frac{\gamma_{rep}}{2z^{12}}.$$

For z<0, $V_{1}^{wall}\left(z\right)=V_{2}^{wall}\left(z\right)=\infty$. We considered $\gamma_{attr}=1,2,3,4,5,6,7,8$ and $\gamma_{rep}=16$ (both in ϵ-units).

The $\left(\rho ,\bar{T}\right)$ phase diagram of this model was determined by theory and MD simulations for $N_{1}=N_{2}$, where $N_{i}$ is the number of particles of the *i*-th component, in [23]. $\bar{T}=kT/\epsilon$, where *k* is the Boltzmann constant and *T* absolute temperature, and $\rho =(N_{1}+N_{2})/V$, with *V* denoting the system volume in $a^{3}$-units. According to the MD results, a dilute gas coexists with a crystalline phase, where alternating bilayers of the first and the second component are formed up to $\bar{T}=0.27$. In each monolayer, particles form a hexagonal pattern. For $\bar{T}>0.27$, alternating bilayers rich in the first and the second component with a liquid structure are formed. At high *T*, a disordered, isotropic phase becomes stable.

In this work, we focus on the adsorption on the selective surface interacting with the particles of the first and the second component with the potentials (Equation 3) and (Equation 4), respectively, at thermodynamic states given by $\bar{T}=0.25$ and the density of gas (away from the adsorbing substrate) $\rho_{g}<0.0027$ (volume fraction $\frac{\pi }{6}\rho_{g}<0.0014$). At this temperature, the gas–crystal coexistence in the bulk occurs for $\rho_{g}=0.0027$. These thermodynamic states correspond to stability of the gas phase not far from the coexistence with the crystal.

### 2.2. The Simulation Method

MD simulations [32] for the system of particles discussed in Section 2.1 were performed in a rectangular box $L_{x}\times L_{y}\times L_{z}$, where $L_{x},L_{y},L_{z}$ are the lengths of the edges. At z=0, the particles interacted with the wall according to Equations (Equation 3) and (Equation 4). At $z=L_{z}$, the reflection boundary condition was imposed for both components via the potential (Equation 4), but with *z* replaced by $L_{z}-z$ and $\gamma_{rep}=1$. In directions *x* and *y* (parallel to the substrate), the periodic boundary conditions were applied.

The equations of motion were solved applying the Verlet algorithm [32] with the time step $\delta t=0.018\tau_{0}$, with the time unit $\tau_{0}=a{\left(m/\epsilon \right)}^{1/2}$, where *m* denotes the particle mass. The temperature was kept constant by scaling the particle velocities once for the time interval $\tau =86-325\tau_{0}$. Because of the adsorption process, the local temperature close to the adsorbing boundary changes much faster than far away from the boundary. To take it into account, the velocity scaling procedure was applied separately in two regions: close to the boundary, i.e., for 0<z<15.0 and away from it for z≥15.0.

The majority of simulations were performed for the initial number of particles $N_{0}=6050$ and $L_{x}=L_{y}=79$. $L_{z}$ was determined by the initial density $\rho_{0}=N_{0}/\left(L_{x}L_{y}L_{z}\right)$: $L_{z}=360.0$ for $\rho_{0}\geq 0.0022$, and $L_{z}=N_{0}/\left(\rho_{0}L_{x}L_{y}\right)$ for $\rho_{0}<0.0022$. For two physically important cases (discussed in Section 3.1), we repeated simulations for 4 times larger systems: $N_{0}=24,200$ and $L_{x}=L_{y}=158$. No physical difference between the results for $N_{0}=6050$ and $N_{0}=24,200$ was noted.

The density of the macroscopically large reservoir does not change during the adsorption process. In simulations performed in a finite system with a fixed number of particles, however, the gas density decreases as a result of the adsorption, especially for dilute gases. In order to mimic the constant-density macroscopic reservoir in MD simulations of a finite system, we keep the density of the gas fixed during the adsorption process by an artificial procedure replacing the inflow of the particles from infinity. Each simulation run started with the nonequilibrium simulation during which the gas particles condensed on the z=0 surface. The fulfillment of the constant gas density condition required a gradual increase in the number of particles so as to compensate for the number of the particles removed from the gas via the adsorption process. The additional particles were inserted at random places far from the adsorbing substrate, i.e., for $l_{z}<z<L_{z}$. The insertion was realized only if the distance between the inserted particle and the surrounding ones was larger than 2.5, otherwise the procedure was repeated. The velocity of the inserted particle was chosen randomly from that of the remaining particles. For $\rho_{0}\geq 0.0022$ we assumed $l_{z}=80$, and $l_{z}$ gradually increased to $l_{z}=150$ for $\rho_{0}$ decreasing to $\rho_{0}=0.0011$. The number of particles to be inserted after a given time interval was taken from the difference between $\rho_{0}$ and the mean density calculated for $l_{z}<z<L_{z}$ and this time interval. When this difference was negative, the particles were not removed from the system. As a consequence, because of the gas density fluctuations, the final value of the gas density $\rho_{g}$ was slightly higher than $\rho_{0}$, but the relative difference never exceeded 1.5%. At the end of the simulation run the system attained the equilibrium state: the total number of particles, *N*, became constant and the system energy fluctuated around the constant value, as is the case in the canonical ensemble. In order to check a possible influence of τ on the simulation results, we continued simulations for τ→∞, i.e., for the microcanonical ensemble (NVE) [32] for two cases. The differences were completely negligible. The total time of the simulation run amounted to 500,000–1,200,000 $\tau_{0}$, and for the equilibrium stage to 66,000–150,000 $\tau_{0}$. All our results presented in Section 3 were obtained in the equilibrium stage. The density profiles were computed in a standard way by averaging over the time interval 16,600 $\tau_{0}$.

## 3. Results

### 3.1. Effect of Wall–Particle Interactions on Distribution of Particles near a Selective Substrate

In this subsection we present distribution of the particles near a selective surface for a fixed temperature $\bar{T}=0.25$ and a fixed gas density, $\rho_{g}=0.0023$. The repulsion strength of the second component is fixed to $\gamma_{rep}=16$, and we compare the effects of moderate, $\gamma_{attr}=4$, with strong, $\gamma_{attr}=6$, attraction of the first-component particles to the substrate. Before describing the adsorption in the mixture, we show as a reference the near-surface structure in the case of the one-component system with particle–particle and wall–particle interactions given by (Equation 1) and (Equation 3), respectively.

#### 3.1.1. The One-Component System with SALR Interactions

Adsorption in the one-component SALR system with the particle–particle and particle–wall interactions (Equation 1) and (Equation 3), respectively, was studied in Ref. [13], but for different thermodynamic states. Here we consider the same temperature $\bar{T}=0.25$ and density $\rho_{1}=0.0023$ as in the binary mixture, to compare adsorption phenomena in the one- and two-component systems in the same conditions. The density profiles in direction perpendicular to the wall averaged over the (x,y) plane and the time interval 16,600 $\tau_{0}$ are shown in Figure 1a,b for $\gamma_{attr}=4$ and $\gamma_{attr}=6$. To show the distribution of the particles in the layer parallel to the substrate with 0<z<1.6, we choose a representative snapshot and present in Figure 1c,d projections on the (x,y) plane of the particles belonging to this layer.

We can see that only a monolayer of particles is adsorbed in both cases. For $\gamma_{attr}=4$, small clusters separated from each other by distances comparable with the range of repulsion are formed. Increasing the attraction strength leads to the formation of elongated clusters, with more than a half of the substrate area empty. The adsorbed monolayer of particles with long-range repulsion screens the attraction of the surface and leads to the formation of a repulsive barrier that inhibits further adsorption of the particles.

#### 3.1.2. The Binary Mixture

In this subsection, we consider the binary mixture with particle–particle interactions (Equation 1) and (Equation 2) between like and different particles, respectively, and wall–particle interactions (Equation 3) and (Equation 4). The density profiles of the two components, averaged over the (x,y) plane and the time interval 16,600 $\tau_{0}$, are shown in Figure 2 as functions of the distance from the wall *z* for $\gamma_{attr}=4$ and $\gamma_{attr}=6$.

We can see that in the mixture the density profile of the first component is significantly different from that in the one-component case, and for the same strength of attraction, four layers of particles instead of one layer are adsorbed. Importantly, particles of the second component are adsorbed between the layers of particles of the first kind. In the one-component case the shape of $\rho_{1}\left(z\right)$ does not change when $\gamma_{attr}$ increases form $\gamma_{attr}=4$ to $\gamma_{attr}=6$; only the maximum increases when the attraction becomes stronger (Figure ). In contrast, in the considered mixture the density profiles for $\gamma_{attr}=4$ and $\gamma_{attr}=6$ have significantly different shapes. In order to understand the origin of the striking difference between the shapes of the density profiles, we examine the projections of the particles in layers of molecular thickness parallel to the substrate on the (x,y) plane.


**Substrate moderately attracting the first component particles.**


Representative configuration for $\gamma_{attr}=4$ is shown in Figure 3. Projections of four layers of particles of thickness comparable with the particle diameter *a* on the (x,y) plane are shown in panels (a)–(d). The simulations show very stable structure in the equilibrium stage, with different snapshots very similar to the one shown in Figure 3. We can see that alternating layers of the first and the second component, perpendicular to the substrate, are formed.

In order to explain why the layered structure, with slabs of the particles perpendicular to the wall, is formed, recall that alternating bilayers of particles of the two components occur in the crystal coexisting with the gas at this temperature and $\rho_{g}=0.0027$. When the repulsion of the second component from the substrate is not very strong, these particles can approach the substrate (see the maximum for z≈1.5 in Figure 2a). The particles of the first kind feel attraction from both the wall and the particles of the second component; therefore, the surface density of these particles can be larger than in the one-component case. Still, the long-range repulsion between particles of the same kind does not allow for full coverage of the surface. The adsorbed stripes of the first-component particles attract particles of the second component, and partially ’screen’ the repulsion from the wall. As a result, a layer of the second-type particles is formed between the layers of particles of the first component. Further away from the surface the wall–particle attraction becomes weaker, and the density as well as order in the subsequent layers decrease. Interestingly, the largest density of the second component is in the central second layer z≈2.5), where there is no repulsion from the wall, and there are still enough attracting particles of the first kind to keep the particles of the second component in the layer between them.


**Substrate strongly attracting the first component particles.**


Let us focus on strong attraction of the particles of the first component, $\gamma_{attr}=6$, with the same strength of the repulsion of the second component from the wall, $\gamma_{rep}=16$ in (Equation 3) and (Equation 4). Projections of the subsequent layers of particles on the (x,y) plane are shown in Figure 4 for a representative configuration. As can be seen by comparing Figure 2a and Figure 3 with Figure 2b and Figure 4, when the substrate is replaced by a strongly attractive one, completely different distribution of the particles occurs. The surface is covered by a dense, homogeneous monolayer of particles of the first component (Figure 2b and Figure 4a), and this layer is followed by an almost empty region of a thickness approximately equal to the particle diameter. After this depletion zone, a dense monolayer of the second component is formed (Figure 2b and Figure 4b). Next to this dense monolayer, clusters of particles of the same component, separated by distances comparable with the range of repulsion, occur (Figure 4c). This structure resembles the adsorbed layer shown in Figure c. In the following layer, particles of the first component form a porous structure, with holes arranged in a local hexagonal pattern (Figure 4d). Some of the holes are empty, but some are filled with clusters of the second component. This structure persists to the next layer of particles, although the density of both components is smaller. In Figure 4e, the double-layer of particles is projected on the (x,y) plane to see the arrangement of the clusters and holes in the *z*-direction. Finally, for 5.9<z<8, excess of the second component can be seen in Figure 2b. As can be seen in Figure 4f, the particles form clusters arranged in a hexagonal pattern for this range of *z*.

Let us explain why the strong attraction of the first component to the wall leads to the radical change of the structure. The long-range repulsion between particles adsorbed at the substrate prevents from close neighborhood of the clusters. However, when this repulsion is overcome by sufficiently strong wall–particle attraction and attraction by the particles of the second component located at a proper distance, the dense monolayer can be energetically favorable. We should stress that the second component plays an important role in stabilizing the monolayer adsorbed at the substrate. The monolayer of the first component ’screens’ the repulsion from the substrate, and a monolayer of the second component is located at a distance from the monolayer of the first component corresponding to the minimum of the potential $u_{12}\left(r\right)$. The monolayer of the second component attracts particles of the same kind at short distances; therefore, a monolayer of clusters is formed in the next layer. On the other hand, the densely packed monolayer of the second component attracts particles of the first component at larger distances, and this leads to a formation of the porous bilayer with some pores filled by clusters of the second kind.

### 3.2. Effect of Gas Density and Surface Selectivity on the Adsorption and Selective Adsorption

The adsorption or coverage by particles of both components is defined by
(5)$$\Gamma =\int_{0}^{\infty }\left(\rho_{1}\left(z\right)+\rho_{2}\left(z\right)-\rho_{g}\right)dz.$$

We also introduce selective adsorption, equal to the surface excess of the concentration, $c\left(z\right)=\rho_{1}\left(z\right)-\rho_{2}\left(z\right)$, by
(6)$$\Gamma_{c}=\int_{0}^{\infty }c\left(z\right)dz.$$

Γ and $\Gamma_{c}$ are shown in Figure 5 as functions of $\rho_{g}$ for $\gamma_{attr}=4$ and $\gamma_{rep}=16$, and as functions of $\gamma_{attr}$ for $\gamma_{rep}=16$, $\rho_{g}=0.0023$ and $\bar{T}=0.25$.

$\Gamma (\rho_{g})$ increases smoothly with increasing $\rho_{g}$ for $\rho_{g}\leq 0.0021$. Between $\rho_{g}=0.0021$ and $\rho_{g}=0.0022$, a substantial increase in the adsorption occurs, and Γ stays almost constant for $\rho_{g}>0.0022$. The selective adsorption is about a factor of 1/3 smaller than Γ for $\rho_{g}\approx 0.0012$. For increasing $\rho_{g}$, $\Gamma_{c}$ increases very slowly, and the difference between Γ and $\Gamma_{c}$ becomes very large for $\rho_{g}=0.0021$. The large increase in Γ for the density increasing from $\rho_{g}=0.0021$ to $\rho_{g}=0.0022$ is accompanied by a visible, but much smaller, increase in $\Gamma_{c}$. This means that, in this mixture with competing interactions, a large adsorption is possible when both components are adsorbed in similar proportions. The long-range attraction between different particles overcomes the long-range repulsion between the particles of the same component. The latter interactions prevented an increase in the adsorption in the one-component system with interactions (Equation 1), as shown in Figure and in Ref. [13].

The change of Γ for $0.0021<\rho_{g}<0.0022$ is associated with a significant change of the near-surface structure. In Figure 6, the densities are shown for $\rho_{g}=0.0021$ and $\gamma_{att}=4$, $\gamma_{rep}=16$. As can be seen by comparing Figure 6 with Figure 2a, the structure near the same substrate changes significantly when the gas density changes from $\rho_{g}=0.0021$ to $\rho_{g}=0.0023$. Two layers of the first component for $\rho_{g}=0.0021$ are replaced by four layers for $\rho_{g}=0.0022$.

## 4. Summary and Discussion

We studied a binary mixture of oppositely charged particles with additional short-range attraction between like particles, and short-range repulsion between different ones near a substrate attracting particles of the first component, and repelling particles of the second one. We found that the distribution of the particles near the wall and the adsorption depend strongly on the strength of the wall–particle interactions and on the gas density. In thermodynamic states not far from the phase transition to a crystalline phase, several layers of particles with various patterns are adsorbed. Two types of significantly different patterns, associated with significantly different adsorption, were found. For moderate attraction, alternating layers of particles of the two components perpendicular to the substrate appear. For strong attraction, alternating layers parallel to the surface, with the two first layers separated by a depletion zone and internal patterns in the subsequent layers appear. In all cases, comparable numbers of particles of the two components are adsorbed. Instead of a porous monolayer in the one-component case, four layers in the mixture with the same attraction of the first component are adsorbed, and for stronger attraction, even for z=7 there is an excess of particles.

Our main conclusions are:The presence of the second component has a huge effect on the adsorption of the first-component particles.The distribution of the particles near the substrate depends qualitatively on the strength of the attraction of the first-component particles to the substrate; the alternating layers of the two component particles are perpendicular to the substrate with moderate attraction (Figure 3), and parallel to the substrate with strong attraction (Figure 4). In the latter case, an ordered internal structure in the parallel layers is present.Despite the repulsion of the second component from the substrate, comparable amounts of the particles of the two components are adsorbed.

Our results were obtained for a particular form of interactions. Now we address the question to what extent these results can be generalized. It is well known that the sequence of ordered phases for increasing density is the same for all one-component SALR systems that can exhibit self-assembly [7]. What depends on the ranges and strengths of the attractive and repulsive parts of interactions is the size of the assemblies and the temperature range of stability of the ordered phases. For the considered mixture, however, the phase diagram was determined only for equal numbers of particles of the two components. What ordered phases appear when the concentration changes is not known even for the bulk system.

To gain some general insight concerning possible patterns in layers of particles parallel to the wall in mixtures with competing interactions, let us consider these layers as quasi-two dimensional (2D) systems. In the mapping of the considered layer on a two-dimensional model, we map the distribution of the particles in the layer on the distribution of projections of the particles on the (x,y) plane. The density in the two-dimensional model is identified with the number of particles in the considered layer per area of the (x,y) plane (in $a^{2}$ units). Examples of the two-dimensional patterns present in our mixture are shown in Figure 3 and Figure 4. In these layers, the density and concentration depend on the distance from the wall and on the wall–particle interactions.

Let us focus on the question about possible patterns in a two-dimensional system that can exhibit spontaneous inhomogeneities on a well-defined length scale. According to the Landau–Brazovskii theory [33], stable structures in a system characterized by a single order-parameter (OP) ϕ are determined by minimums of the functional
(7)$$\begin{array}{cc} L_{B}\left[\phi \right]=\int dr & \left[f\left(\phi \left(r\right)\right)+\frac{\beta V_{2}}{2}{\left|\nabla \phi \left(r\right)\right|}^{2}+\frac{\beta V_{4}}{4!}{\left(\nabla^{2}\phi \left(r\right)\right)}^{2}\right], \end{array}$$
where
(8)$$f\left(\phi \right)=\left(A_{2}/2+\beta V_{0}\right)\phi^{2}+A_{3}\phi^{3}/3!+A_{4}\phi^{4}/4!,$$

$V_{4}>0$ and $A_{2},A_{4}>0$. The parameters $A_{n}$ are associated with entropy, and the parameters $V_{n}$ are associated with the interaction potentials. The inhomogeneous structure (oscillatory ϕ(r)) is favored by the second term in Equation (Equation 7) if $V_{2}<0$ ( if $V_{2}<0$, $L_{B}$ decreases for $|\nabla \phi \left(r\right)|$ increasing from zero), and the length-scale of inhomogeneities is $-\left(\pi /6\right)\left(V_{4}/V_{2}\right)$ [33]. At sufficiently low *T*, ordered, periodic patterns are predicted by the Brazovskii functional. In 2D, parallel stripes with alternating positive and negative values of ϕ occur for small values of $|A_{3}|$. For intermediate values of $|A_{3}|$, circular domains forming a hexagonal pattern appear. Inside the circles, ϕ(r)>0 if $A_{3}<0$, and ϕ(r)<0 if $A_{3}>0$. For very large $|A_{3}|$, the homogeneous structure, with ϕ>0 for $A_{3}>0$, and ϕ<0 for $A_{3}<0$ is stable.

In the one-component SALR systems, the OP is a local deviation of density from the space-averaged value. The functional (Equation 7) was obtained from the grand-potential functional in kT=1/β units, βΩ[ρ], by a coarse-graining procedure in [7]. It was shown that $V_{2}<0$ for sufficiently strong long-range repulsion, and $A_{n}$ is equal to the *n*-th derivative with respect to ρ of the entropic contribution to βΩ[ρ]. With the assumption of the lattice-gas entropy, $A_{3}={\left(1-\rho \right)}^{-2}-\rho^{-2}$. This form of $A_{3}$ is oversimplified on the quantitative level, but it easily shows qualitative trends. In a one-component system, we should observe dilute gas, clusters, stripes, voids and a dense liquid for increasing density ρ. The same sequence of phases was obtained for the monolayer of identical particles interacting with the sum of the Casimir and the dipole–dipole potentials in [27] by the density functional theory.

For a liquid mixture with fixed density, the OP is a local deviation of the concentration *c* from its average value, and ϕ>0 (ϕ<0) means excess of particles of the first (second) component. For a symmetrical binary mixture with fixed density ρ, the functional (Equation 7) can be obtained in a similar way as in [7], and $A_{n}$ is equal to the *n*-th derivative of the entropic contribution to βΩ[ρ,c] with respect to *c*. The simplest lattice-gas entropy leads to $A_{3}=\left[{\left(\rho -c\right)}^{-2}-{\left(\rho +c\right)}^{-2}\right]/2$. Thus, for relatively large |c| circular domains of the minority component in the background of the majority component should occur, and alternating stripes of the first and the second component should be formed for relatively small difference in the number of particles of the two components.

As we can see from the density profiles (Figure 2), there are layers that can be treated as a one-component system, and layers with different proportions of the two components. In the first case ϕ(x,y) should be interpreted as the local excess of density, and in the second case as the local excess of concentration.

The patterns shown in Figure 3 and Figure 4 agree very well with the general predictions discussed above. Thus, we may expect similar patterns in various systems with this type of interactions, except that the size of the clusters or layers will be determined by the range of the attractive and repulsive parts of the potential, and the strength of the interactions determines the temperature range for which structures with long- or short-range order can appear [7]. However, we still need a theory capable to predict the density and concentration profiles averaged over the (x,y) plane, ρ(z) and c(z) near surfaces with different wall–particle interactions.

Our results show that adsorption phenomena in self-assembling mixtures can be very rich. Anomalous adsorption in the one-component SALR system is replaced by formation of a thick adsorbed layer with internal regular patterns depending on the type of the substrate and the gas density.

## Figures and Tables

**Figure 1 molecules-26-04532-f001:**
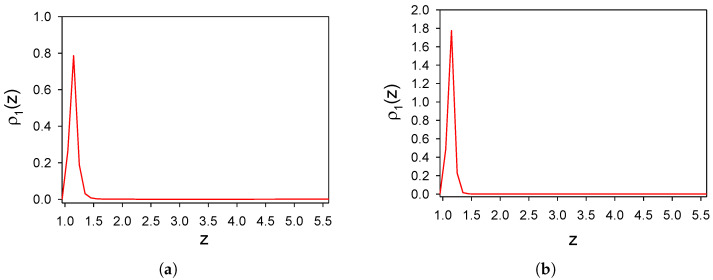
Distribution of the particles near an attractive surface in the one-component system with particle–particle and wall–particle interactions given by (Equation 1) and (Equation 3), respectively, for $\bar{T}=0.25$ and $\rho_{1}=0.0023$ (volume fraction $\frac{\pi }{6}\rho_{1}=0.0012$). Dimensionless density $\rho_{1}\left(z\right)$ as a function of a distance *z* from the surface is shown for $\gamma_{attr}=4$ and $\gamma_{attr}=6$ in panels (**a**,**b**), respectively. Projection on the (x,y) plane of particles (red circles) belonging to a layer of molecular thickness parallel to this plane, 0<z<1.6, is shown for $\gamma_{attr}=4$ and $\gamma_{attr}=6$ in panels (**c**,**d**), respectively, for representative configurations (snapshots). X,Y and *z* are in units of the diameter of the particles *a*. Simulations were performed for the initial number of particles $N_{0}=6050$.

**Figure 2 molecules-26-04532-f002:**
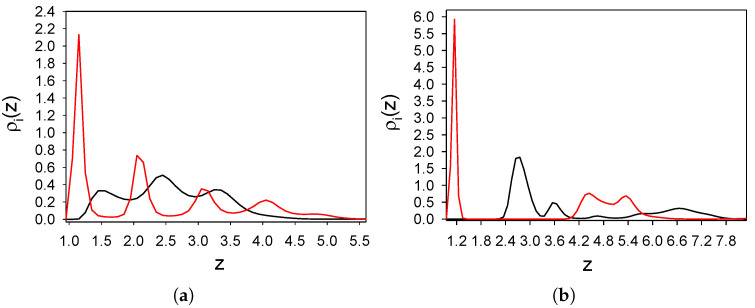
Distribution of the particles near a surface attracting the first and repulsing the second component for $\bar{T}=0.25$ and $\rho_{g}=0.0023$ (volume fraction $\frac{\pi }{6}\rho_{1}=0.0012$). Particle–particle interactions are given by (Equation 1) and (Equation 2) for like and different particles, and wall–particle interactions are given by (Equation 3) and (Equation 4) for the first and the second component, respectively. Dimensionless densities $\rho_{1}\left(z\right)$ (red line) and $\rho_{2}\left(z\right)$ (black line) of the first and the second component as functions of a distance *z* from the surface are shown for (**a**) $\gamma_{attr}=4$ and (**b**) $\gamma_{attr}=6$. In both panels $\gamma_{rep}=16$. $\bar{T}=kT/\epsilon$, and $\rho_{g}$ denotes the dimensionless gas density far from the adsorbing substrate. *z* is in units of the particle diameter *a*. Simulations were performed for the initial number of particles $N_{0}=6050$. Very similar results were obtained for $N_{0}$ = 24,200.

**Figure 3 molecules-26-04532-f003:**
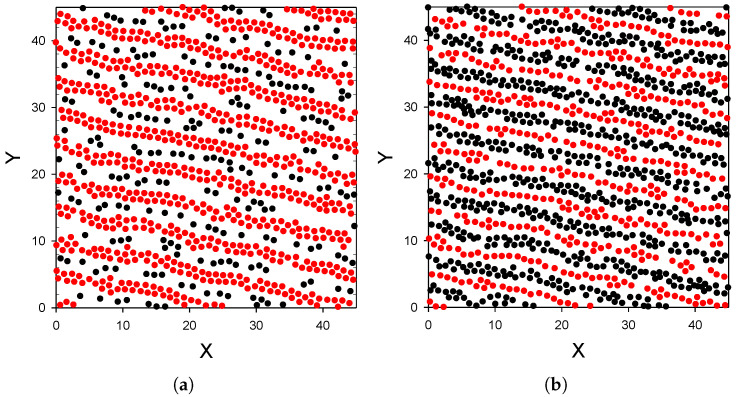
Distribution of the particles near a surface attracting the first and repulsing the second component for $\bar{T}=0.25$ and $\rho_{g}=0.0023$ (volume fraction $\frac{\pi }{6}\rho_{1}=0.0012$). Particle–particle interactions are given by (Equation 1) and (Equation 2) for like and different particles, and wall–particle interactions are given by (Equation 3) and (Equation 4) for the first and the second component, respectively, with $\gamma_{attr}=4$ and $\gamma_{rep}=16$. Projections on the (x,y) plane of the particles in the layers 0<z<1.6, 1.6<z<2.6, 2.6<z<3.6 and 3.6<z<4.6 are shown in panels (**a**–**d**), respectively, for a representative configuration. Red and black circles represent particles of the first and the second kind, respectively. X,Y and *z* are in units of the particle diameter *a*. Only a part of the simulation box is shown. Simulations were performed for the initial number of particles $N_{0}=6050$. Very similar results were obtained for $N_{0}$ = 24,200.

**Figure 4 molecules-26-04532-f004:**
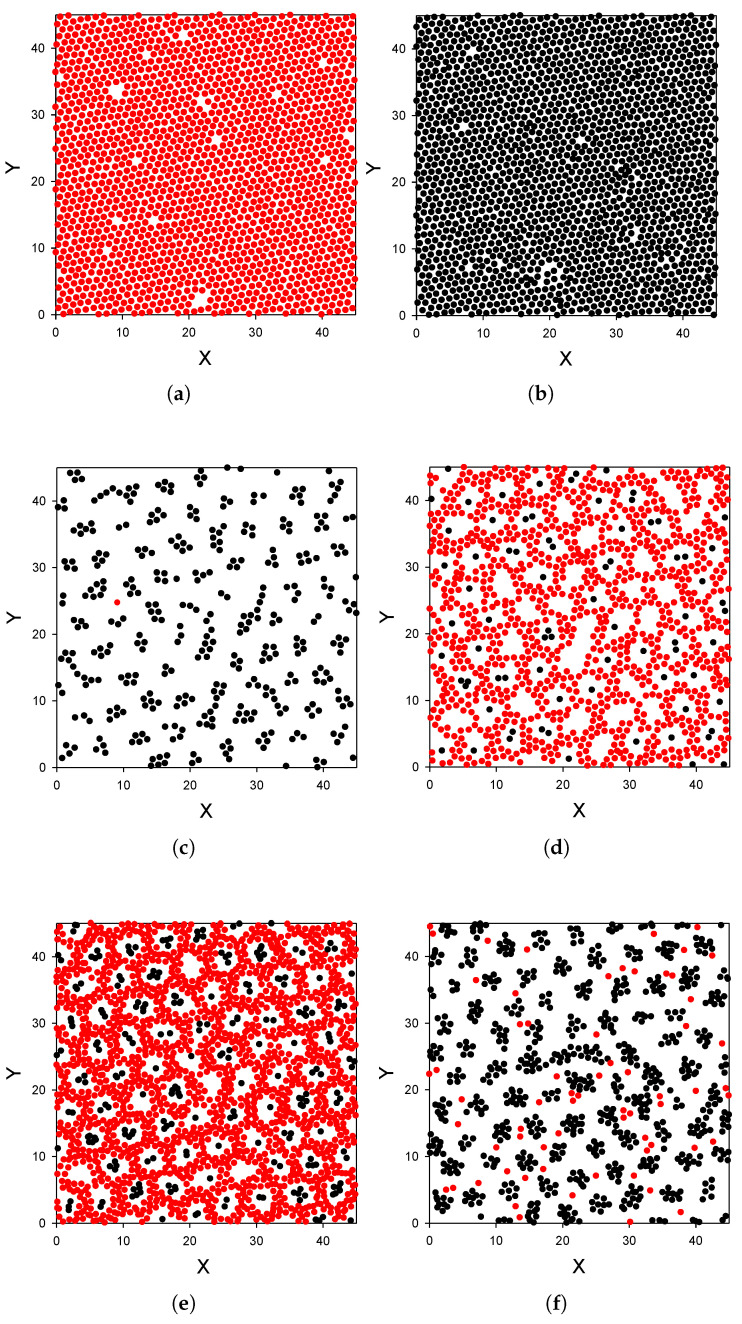
Distribution of the particles near a surface strongly attracting the first, and moderately repelling the second component. In panels (**a**–**d**) projections on the (x,y) plane of the particles in the monolayers 0.9<z<1.5, 2.3<z<3.3, 3.3<z<4.0 and 4.0<z<5.0 are shown for a representative configuration. Almost no particles are present for 1.5<z<2.3, i.e., between panels (**a**,**b**). In panels (**e**,**f**), projections of particles in double layers of thickness ∼2, with 4.0<z<5.9 and 5.9<z<8, respectively, are shown. By comparing (**d**,**e**), one can see the particle distribution in the *z*-direction. In the case of the double layer, projections of the particles can overlap. Red and black circles represent particles of the first and the second kind, respectively. The particles of the first and the second component interact with the wall at z=0 with (Equation 3) and (Equation 4) with $\gamma_{attr}=6$ and $\gamma_{rep}=16$, respectively. X,Y and *z* are in units of the particle diameter *a*. $\bar{T}=0.25$ and $\rho_{g}=0.0023$ (volume fraction $\frac{\pi }{6}\rho_{1}=0.0012$). Simulations were performed for the initial number of particles $N_{0}=6050$. Very similar results were obtained for $N_{0}$ = 24,200.

**Figure 5 molecules-26-04532-f005:**
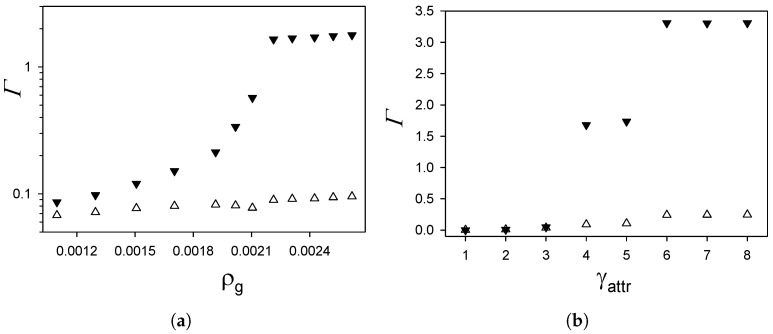
The adsorption Γ (filled symbols) and the selective adsorption $\Gamma_{c}$ (open symbols) (Equation 5) and (Equation 6), respectively, in units of $1/a^{2}$, where *a* is the particle diameter, as functions of dimensionless gas density $\rho_{g}$ for $\gamma_{attr}=4$ and $\gamma_{rep}=16$ (**a**) and as functions of the strength of the attractive interactions between the substrate and the first component $\gamma_{attr}$ for $\gamma_{rep}=16$ and $\rho_{g}=0.0023$ (**b**). $\bar{T}=0.25$ in both panels. Simulations were performed for the initial number of particles $N_{0}=6050$.

**Figure 6 molecules-26-04532-f006:**
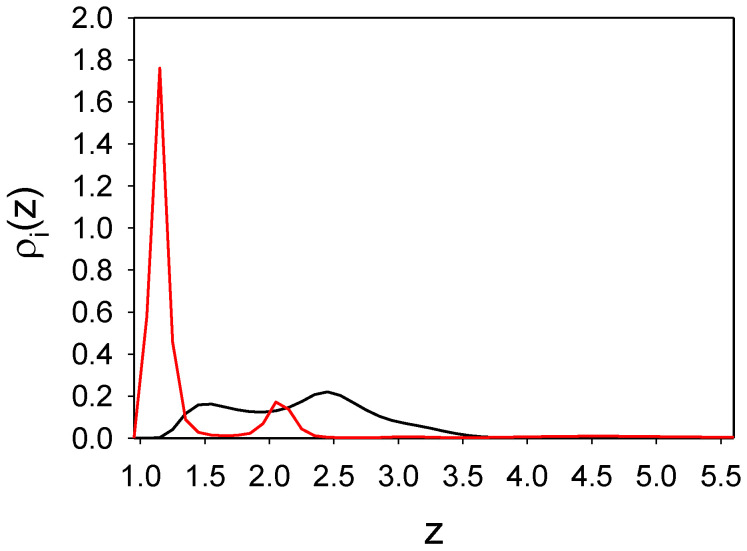
Adsorption at a surface for $\bar{T}=0.25$ and $\rho_{g}=0.0021$. Red and black lines represent average densities of the first and the second component, $\rho_{1}\left(z\right)$ and $\rho_{2}\left(z\right)$, respectively, as functions of a distance *z* from the surface. The first and the second component interact with the wall with (Equation 3) and $\gamma_{attr}=4$, and with (Equation 4) and $\gamma_{rep}=16$, respectively. Simulations were performed for the initial number of particles $N_{0}=6050$.

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
