# Peer review of "Adsorption in Mixtures with Competing Interactions"

_molecules, 2021, doi:10.3390/molecules26154532_

Round 1

Reviewer 1 Report

In this manuscript the authors study the behavior of a binary mixture of

particles with competing interactions close to a wall. Systems with competing interactions have been studied in various situations with different methods

and it is found that these systems can show very interesting and rich phase behavior without confining potential. Here the system considered is enriched by (1) considering a binary mixture and (2) by adding a planar wall that attracts one and repels the second component. The bulk behavior has been studied by the authors before.

The system is interesting and the behavior is rich. One can find some similarities with one component systems with competing interactions, but as the authors show clearly, there are additional phenomena that can be found only in this particular mixture.

There are a few things that need to be clarified.

The author mention in the introduction that the attraction in case of a competing interaction could be due to critical Casimir interaction. I think the authors should point out that this has been employed in recent DFT studies for a two dimensional system with competing interaction that was constructed so that it should be accessible in experiments. See Marolt et al. Phys Rev. E 100, 052602 (2019) and Phys. Rev. E 102, 042608 (2020).

The authors introduce the particle-particle interaction potentials employed in the present study as if they can be realized in experiments. Is this the case or should a word of warning, due to the findings of Ref. 8, be added that this is a theoretical model?

The wall potential is not entirely clear to me. Is the wall potential infinite for z<=0? In that case the authors should write that a hard wall potential is used for z<=0.

In line 115 the authors mention that the temperature is kept constant by scaling the velocities in a time interval. I do not understand at all what is meant by this sentence. Also the meaning of tau = ... is not clear. Please spell out the meaning.

In the manuscript the authors write that they start with a certain number of particles and that sometimes new particles are inserted. Usually a molecular dynamics simulation is performed with a fixed number of particles. How should one think about this simulation. Are the performed in the canonical or grand canonical ensemble or is this something different altogether?

The results shown in Figs. 1-4 and 6 need to be explained a little bit more. For example in Fig.1 (a) and (b) the density profile seem to be averaged profiles. Is this correct? If so are the time averaged? Over which time interval? The plots in Fig.1 (c) and (d) seem to be projections of snapshots. Is this correct? Are the projections done by projecting all particles in a given z interval onto the x-y plane? Can particle overlap in such a projection happen? If it is from a snapshot, how is it chosen? If it is not from a snapshot, how is it done? Similar questions arise with Figs. 2, 3, 4, and 6.

Also the projections for different z-intervals are called adsorption at the wall. Would it be clearer to call it cuts through the density profile or snapshot?

In the summary starting around line 290 the authors make connection to two dimensional systems with competing interactions. I think it would be interesting to re-connect to the work by Marolt et al. mentioned in connection with the Casimir interaction earlier in this report.

Once these points have taken into account by the authors, this manuscript can be considered for publication in Molecules.

Author Response

Referee 1

In this manuscript the authors study the behavior of a binary mixture of particles with competing interactions close to a wall. Systems with competing interactions have been studied in various situations with different methods and it is found that these systems can show very interesting and rich phase behavior without confining potential. Here the system considered is enriched by (1) considering a binary mixture and (2) by adding a planar wall that attracts one and repels the second component. The bulk behavior has been studied by the authors before. The system is interesting and the behavior is rich. One can find some similarities with one component systems with competing interactions, but as the authors show clearly, there are additional phenomena that can be found only in this particular mixture.

There are a few things that need to be clarified.

The author mention in the introduction that the attraction in case of a competing interaction could be due to critical Casimir interaction. I think the authors should point out that this has been employed in recent DFT studies for a two dimensional system with competing interaction that was constructed so that it should be accessible in experiments. See Marolt et al. Phys Rev. E 100, 052602 (2019) and

Phys. Rev. E 102, 042608 (2020).

The authors introduce the particle-particle interaction potentials employed in the present study as if they can be realized in experiments. Is this the case or should a word of warning, due to the findings of Ref. 8, be added that this is a theoretical model?

our response

In the revised manuscript we add the required info and citations in the third paragraph on page 2, lines 73-87 (in red font).

The wall potential is not entirely clear to me. Is the wall potential infinite for z<=0? In that case the authors should write that a hard wall potential is used for z<=0.

our response

In the revised manuscript we add the required info above equation (3) and below equation (4) (in red font).

In line 115 the authors mention that the temperature is kept constant by scaling the velocities in a time interval. I do not understand at all what is meant by this sentence. Also the meaning of tau = ... is not clear. Please spell out the meaning. In the manuscript the authors write that they start with a certain number of particles and that sometimes new particles are inserted. Usually a molecular dynamics simulation is performed with a fixed number of particles. How should one think about this simulation. Are the performed in the canonical or grand canonical ensemble or is this something different altogether?

our response

In the revised manuscript we extend the description of the simulation procedure. The additions in sec.2.2 are in red font.

The results shown in Figs. 1-4 and 6 need to be explained a little bit more. For example in Fig.1 (a) and (b) the density profile seem to be averaged profiles. Is this correct? If so are the time averaged? Over which time interval? The plots in Fig.1 (c) and (d) seem to be projections of snapshots. Is this correct? Are the projections done by projecting all particles in a given z interval onto the x-y plane? Can particle

overlap in such a projection happen? If it is from a snapshot, how is it chosen? If it is not from a snapshot, how is it done? Similar questions arise with Figs. 2, 3, 4, and 6.

our response

In the revised manuscript we provide the required info in red font. At the end of sec.2.2 we write how the densities were computed and over which time interval. In sec.3.1 we write explicitly both in the text and in figure captions that the plots in Fig.1 (c) and (d) and Figs.2,3,4 and 6 are projections of representative snapshots. We add the info that we observed very stable structure, with snapshots very similar to one another (line 238). In Fig. 4 caption we write (in red font) that the projections of all particles belonging to a double layer can overlap.

Also the projections for different z-intervals are called adsorption at the wall. Would it be clearer to call it cuts through the density profile or snapshot?

our response

In the revised manuscript we change the figure captions by replacing ‘adsorption at the wall’ by ‘Distribution of the particles near a surface’

In the summary starting around line 290 the authors make connection to two dimensional systems with competing interactions. I think it would be interesting to re-connect to the work by Marolt et al. mentioned in connection with the Casimir interaction earlier in this report.

our response

In the revised manuscript we add a comment and the citation in line 383 (in red font).

Once these points have taken into account by the authors, this manuscript can be considered for publication in Molecules.

Reviewer 2 Report

The authors studied a binary mixture of oppositely charged particles with additional short range attraction between like particles, and short-range repulsion between different ones near a substrate attracting particles of the first component, and repelling particles of the second one. The presentation of methods and scientific results are satisfactory for publication in the Molecules journal. The minor and significant drawbacks to be addressed can be specified as follows:

  1. Information about the adsorbent has been missed. What is the type: amorphous or ideal solid? Virtual adsorbent? What is the relationship of the studied solid with real adsorbents (metals (eg. Cu(1 1 1), Cu (1 1 0) Cu (1 0 0)), graphite, salts, etc.)? Can the choice of the substrate affect the obtained results? Would you mind providing any details?
  2. Time step? Thermostat? Would you please provide any details?
  3. 2 and subsequent figures, see figure captions. „Simulations were performed for the initial number of particles N0 = 6050 and N0 = 24200, with almost the same results in the two cases.” Does this figure (i.e. Fig. 2 ) show the results for 6050 and 24200? Are the same values in these plots?

Author Response

Referee 2

The authors studied a binary mixture of oppositely charged particles with additional short range attraction between like particles, and short-range repulsion between different ones near a substrate attracting particles of the first component, and repelling particles of the second one. The presentation of methods and scientific results are satisfactory for publication in the Molecules journal. The minor and

significant drawbacks to be addressed can be specified as follows:

1. Information about the adsorbent has been missed. What is the type: amorphous or ideal solid? Virtual adsorbent? What is the relationship of the studied solid with real adsorbents (metals (eg. Cu(1 1 1), Cu (1 1 0) Cu (1 0 0)), graphite, salts, etc.)? Can the choice of the substrate affect the obtained results? Would you mind providing any details?

our response

In the revised manuscript we add the info about the adsorbent at the beginning of the second paragraph in sec. 2.1 (in red font). We write that the adsorbent is ideal and explain why we make such an assumption.

2. Time step? Thermostat? Would you please provide any details?

our response

In the revised manuscript we extend the description of the simulation method in sec. 2.2 and add the required info. The additions are in red font.

3. 2 and subsequent figures, see figure captions. „Simulations were performed for the initial number of particles N0 = 6050 and N0 = 24200, with almost the same results in the two cases.” Does this figure (i.e. Fig. 2 ) show the results for 6050 and 24200? Are the same values in these plots?

our response

In the revised manuscript we modified the figure captions to make it clear that the presented results are for 6050 particles, and that they remain very similar for 24200 particles. The modified text is in red font.

Reviewer 3 Report

The manuscript entitled "Adsorption in mixtures with competing interactions" presented a research paper related to adsorption in a mixture of particles with forming competing interactions. The main issues related to the paper refer to the author's presentation of results. Also, the scientific contribution and novelty of this work have been not highlighted. Too many results are presented without organization in the manuscript. Obtained results should be used for the investigation of the mechanism of adsorption. Otherwise, obtained data did not useful. Also, the title of the paper should be revised. I do not recommend the acceptance of this paper in its current state.

Author Response

Referee 3

The manuscript entitled "Adsorption in mixtures with competing interactions"presented a research paper related to adsorption in a mixture of particles with forming competing interactions. The main issues related to the paper refer to the author's presentation of results. Also, the scientific contribution and novelty of this work have been not highlighted.

our response

In the revised manuscript we add the info about the novelty of the considered questions at the end of the second and third paragraph in the introduction (in red font).

Too many results are presented without organization in the manuscript. Obtained results should be used for the investigation of the mechanism of adsorption. Otherwise, obtained data did not useful.

our response

In the revised manuscript we separate the description of the results in sec. 3.1.2 into two parts, and add titles “Substrate moderately attracting the first component particles.” for the first part and “Substrate strongly attracting the first component particles” for the second part. We hope this improves the organization of the presentation of the results. In addition, we add a paragraph (second paragraph in sec.4 in the revised manuscript) with a list of conclusions (in red font).

Also, the title of the paper should be revised. I do not recommend the acceptance of this paper in its current state.

our response

We think that the title is appropriate. The other referees did comment on the title.

Reviewer 4 Report

By molecular dynamics simulations, the authors investigate self-assembled mixtures of oppositely charged particles in the light of their adsorption onto a substrate. The text is mostly clear and concise making it suitable for publication in a special issue about molecular adsorption in Molecules. Hence, I support its publication with minor revisions that should be included in the new version of the manuscript, as stated below:

1. Does the proposed model take into account the softness or roughness of the particles? Could the obtained results be extended to, for example, microgel (soft) and metal (hard) particles?

2. What is the volume fraction of the particles used in the studies? This is important information that is not very clear in the current version of the manuscript.

3. Extensive experimental results from the literature show that several systems composed by particle assembly form cubic crystals instead of alternating planar layers. Is there any experimental result confirming what has been found in the simulations?

Author Response

Referee 4

By molecular dynamics simulations, the authors investigate self-assembled mixtures of oppositely charged particles in the light of their adsorption onto a substrate. The text is mostly clear and concise making it suitable for publication in a special issue about molecular adsorption in Molecules. Hence, I support its publication with minor revisions that should be included in the new version of the manuscript, as stated below:

1. Does the proposed model take into account the softness or roughness of the particles? Could the obtained results be extended to, for example, microgel (soft) and metal (hard) particles?

our response

In the revised manuscript we add the required info at the end of the first paragraph in sec. 2.1, in red font.

2. What is the volume fraction of the particles used in the studies? This is important information that is not very clear in the current version of the manuscript.

our response

In the revised manuscript we provide the info about the volume fraction of the particles in the gas in a more visible way both in the text (line 139, red font) and in figure captions.

3. Extensive experimental results from the literature show that several systems composed by particle assembly form cubic crystals instead of alternating planar layers. Is there any experimental result confirming what has been found in the simulations?

our response

In the revised manuscript we add the required info at the end of the first paragraph on page 3 (starting at line 97, in red font).

Round 2

Reviewer 3 Report

The authors have been made many corrections in accordance with review suggestions. The manuscript could be accepted.